
# Vertical characterization of aerosol optical properties and brown carbon in winter in urban Beijing, China

Conghui Xie[1,2], Weiqi Xu[1,2], Junfeng Wang[4], Qingqing Wang[1], Dantong Liu[5], Guiqian Tang[1], Ping Chen[6], Wei Du[1,2], Jian Zhao[1,2], Yingjie Zhang[1], Wei Zhou[1,2], Tingting Han[1], Qingyun Bian[2,7], Jie Li[1], Pingqing Fu[1,2], Zifa Wang[1,2], Xinlei Ge[4], James Allan[5,8], Hugh Coe[5], Yele Sun[1,2,3]

[1]State Key Laboratory of Atmospheric Boundary Layer Physics and Atmospheric Chemistry, Institute of Atmospheric Physics, Chinese Academy of Sciences, Beijing 100029, China

[2]College of Earth Sciences, University of Chinese Academy of Sciences, Beijing 100049, China

[3]Center for Excellence in Regional Atmospheric Environment, Institute of Urban Environment, Chinese Academy of Sciences, Xiamen 361021, China

[4]School of Environmental Science and Engineering, Nanjing University of Information Science & Technology, Nanjing 210044, China

[5]Centre for Atmospheric Science, School of Earth, Atmospheric and Environmental Science, University of Manchester, Manchester M13 9PL, UK

[6]Handix Scientific LLC, Boulder, CO 80301, USA

[7]CAS Key Laboratory of Regional Climate-Environment Research for Temperate East Asia, Institute of Atmospheric Physics, Chinese Academy of Sciences, Beijing 100029, China

[8]National Centre for Atmospheric Science, The University of Manchester, Manchester, UK

*Correspondence*: Yele Sun (sunyele@mail.iap.ac.cn)

**Abstract.** Aerosol particles are of importance in the Earth's radiation budget since they scatter and absorb sunlight. While extensive studies of aerosol optical properties have been conducted at ground sites, vertical measurements and characterization are very limited in megacities. In this work, we present simultaneous real-time online measurements of aerosol optical properties at ground level and at 260 m on a meteorological tower from 16 November to 13 December in 2016 in Beijing along with measurements of continuous vertical profiles during two haze episodes. The average (±1σ) scattering and absorption coefficients ($b_{sca}$ and $b_{abs}$, λ = 630 nm) were 337.6 (±356.0) and 36.6 (±33.9) Mm$^{-1}$ at 260 m, which were 26.5% and 22.5% lower than those at ground level. Single scattering albedo (SSA), however, was comparable between the two heights with slightly higher values at ground level (0.89 ±0.04). Although $b_{sca}$ and $b_{abs}$ showed overall similar temporal variations between ground and 260 m, the ratios of 260 m to ground

varied substantially from less than 0.4 during the cleanest stages of haze episodes to > 0.8 in the late afternoon. A more detailed analysis indicates that vertical profiles of $b_{sca}$, $b_{abs}$, and SSA in the low atmosphere were closely related to the changes in meteorological conditions and mixing layer height. The mass absorption cross-section MAC of BC ($\lambda = 630$ nm) varied substantially from 9.5 to 13.2 $m^2\,g^{-1}$ in winter in Beijing, and it was strongly associated with the mass ratio of non-refractory BC ($r$BC) materials

to $r$BC ($M_R$), and also the oxidation degree of organics in $r$BC-containing particles. Our results show that the increases in MAC of BC in winter were mainly caused by photochemically produced secondary materials. Light absorption of organic carbon (brown carbon, BrC) was also important in winter, which on average accounted for 46 (±8.5) % and 48 (±9.3) % of the total absorption at 370 nm at ground level and 260 m, respectively. A Linear regression model combined with positive matrix factorization analysis was used to show that coal combustion was the dominant source contribution of BrC (48-55%) followed by biomass burning (17%)

and photochemically processed secondary organic aerosol (~20%) in winter in Beijing.

## 1 Introduction

Light scattering and absorption of aerosols reduce visibility and affect the radiation and energy budget of the Earth (Rosenfeld, 1999;Kim and Ramanathan, 2008). Scattering aerosols cool atmosphere and exert a negative forcing while light absorbing materials warm the atmosphere and, if they exist over a brighter underlying surface, contribute a positive forcing (Haywood and Boucher,

2000). While scattering particles  mainly include ammonium sulfate, ammonium nitrate, and a majority of organics (Han et al., 2015), black carbon (BC) and brown carbon (BrC) are the two major light absorbing matters in fine particles (Bond et al., 2013). Although numerous studies have been conducted to investigate aerosol optical properties, accurate quantification of aerosol radiative forcing still remains challenging (Stocker et al., 2013). While the very complex optical properties affected by mixing states are one of the reasons (Peng et al., 2016;Cappa et al., 2012;Liu et al., 2017), our limited understanding of the vertical distributions of optical

properties and their relationship to composition and mixing state is also important. For example, a recent study found that BC near the capping inversion is more effective in suppressing planetary boundary layer height and weakening turbulent mixing (Wang et al., 2018;Ding et al., 2016).



Most previous measurements of aerosol particles in urban Beijing are conducted at ground level (Sun et al., 2013;Sun et al., 2014;Tao et al., 2015), which are subject to the influences of local emissions, such as biomass burning, cooking and traffic (Sun et al., 2014). Until recently, Sun et al. (2015) reported the first simultaneous measurements of PM$_1$ composition at 260 m and ground level in urban Beijing in winter 2013. Higher nitrate and much lower primary organic aerosol (POA) at 260 m than ground level was

observed. This result is consistent with the subsequent observations in autumn 2015 (Zhao et al., 2017). The size distributions of aerosol at 260 m and ground level are also different especially for Aitken mode particles (Du et al., 2017). Such differences in concentration, composition and size distributions between 260 m and ground level may exert a potential influence on aerosol optical properties. However, the vertical distributions of aerosol optical properties are rarely characterized. Wang et al. (2017b) conducted the first continuous vertical measurements of particle extinction and BC from ground to 200-260 m during two severe haze episodes

in winter. The results showed four distinct types of vertical profiles that were closely linked with boundary layer dynamics. However, this study is limited within a relatively short time, and the real-time measurements of aerosol optical properties at different heights in Beijing is yet to be performed.

Black carbon, as the major light absorbing matter in atmospheric aerosol (Bond et al., 2013), absorbs visible light strongly with an absorption Angstrom exponent (AAE) of ~1 (Horvath, 1997). The light absorbing ability of BC is affected by the coating materials which may change substantially during the aging process. The enhancement of BC absorption due to non-BC materials in BC-

containing particles is often referred to as lensing effect (Bond et al., 2006). Previous studies found that internally mixed particulate organic matter with BC could enhance absorption by up to 70% (Lack et al., 2012), and the absorption enhancement strongly depends on the BC coating amount and particle mixing state (Liu et al., 2017;Liu et al., 2015b). Although Peng et al. (2016) found that the BC light absorption enhancement can be significantly increased in polluted urban environment by using a novel environmental

chamber approach, the evaluation of such effects through field measurements are still limited in Beijing. In addition to BC, light-absorbing organic carbon (brown carbon, BrC) plays an important role in affecting radiative forcing at ultraviolet wavelength (Laskin et al., 2015) although its mass absorption cross-section (MAC, also known as mass absorption efficiency) is order of magnitude less than that of BC in the visible wavelength (Yang et al., 2009). The AAE of BrC is ubiquitously larger than 1, yet

accurate quantification of absorption of BrC is challenging (Laskin et al., 2015), particularly in a megacity with complex source emissions and processes. For example, recent studies have found that coal combustion is also an important source of BrC in winter in Beijing in addition to biomass burning aerosol (Sun et al., 2017;Yan et al., 2017).

In this work, we conducted comprehensive measurements of aerosol optical properties including light extinction coefficients ($b_{ext}$), scattering coefficients ($b_{sca}$), and absorption coefficients ($b_{abs}$) at both ground level and 260 m on a meteorological tower in winter in 2016 along with two continuous vertical measurements of $b_{sca}$ and $b_{abs}$ between ground level and 260 m. The measurements of aerosol optical properties from different instruments are compared, and the vertical differences in $b_{sca}$, $b_{abs}$, and single scattering albedo (SSA) are characterized. Also, the evolution of vertical differences in aerosol optical properties and its relationship with meteorological conditions and aerosol composition are illustrated. In addition, the light absorption capability of BC is estimated, and the contribution of BrC to $b_{abs}$ at 370 nm are quantified.

## 2. Experimental methods

### 2.1 Sampling site and measurements

All measurements at ground level and 260 m were conducted at the tower site of Institute of Atmospheric Physics (IAP), Chinese Academy of Sciences (39°58′28″N, 116°22′16″E) in Beijing from 16 November to 13 December 2016 as a part of APHH-Beijing winter campaign. A detailed description of the sampling site is given in Zhou et al. (2018).

Table 1 lists a summary of measurements in this study. At the ground site, $b_{ext}$ and $b_{sca}$ (λ=630 nm) of dry particles (PM$_1$) were measured by a cavity attenuated phase shift single scattering albedo monitor (CAPS PM$_{ssa}$, Aerodyne Research Inc.) (Onasch et al., 2015) that was installed in a room on the rooftop of a two-story building (~ 8 m). Note that the major uncertainty in scattering measurements caused by truncation effect is less than 2% (Han et al., 2017). The detailed description of the CAPS PM$_{ssa}$ is given in Han et al. (2017). In the same room, a seven-wavelength (370, 470, 520, 590, 660, 880, and 950 nm) Aethalometer (AE33, Magee Scientific Corp.) was used to measure BC at a time resolution of 1 min. This version of the Aethalometer uses a compensation



algorithm based on "dual-spot" measurements to automatically correct the filter-based loading effects (Drinovec et al., 2015). In addition, a photoacoustic extinctiometer (PAX, Droplet Measurement Technologies) was used to measure $b_{sca}$ and $b_{abs}$ ($\lambda$=870 nm) of dry $PM_{2.5}$ in a container at ground level (~3 m). Other collocated measurements at the same ground site include non-refractory submicron (NR-$PM_1$) aerosol species (organics, sulfate, nitrate, ammonium, and chloride) by an Aerodyne high-resolution time-of-flight aerosol mass spectrometer (HR-AMS hereafter) and refractory black carbon ($r$BC) and BC-containing species by a soot particle aerosol mass spectrometer (SP-AMS)(Wang et al., 2017a;Onasch et al., 2012). In this study, the tungsten vaporizer of the SP-AMS was removed, thus only $r$BC-containing particles were measured. Positive matrix factorization (PMF) was performed on the high-resolution mass spectra of organic aerosol (OA) of the HR-AMS and the SP-AMS. Six OA factors were identified from HR-AMS measurements including fossil-fuel-related OA (FFOA) dominantly from coal combustion, cooking OA (COA), biomass burning OA (BBOA), oxidized POA (OPOA), oxygenated OA (OOA), and aqueous-phase OOA (aq-OOA) (Xu et al., 2018). In comparison, four OA factors were identified from SP-AMS measurements including FFOA, BBOA, OOA1 and OOA2. More detailed descriptions of the operations and calibrations of the HR-AMS and SP-AMS, and subsequent PMF analysis can be found in Xu et al. (2015) and Wang et al. (2016).

At 260 m on the Beijing 325 m Meteorological Tower (BMT), $b_{ext}$ ($\lambda$=630 nm) of $PM_{2.5}$ was measured by a cavity attenuated phase shift extinction monitor (CAPS $PM_{ext}$ Aerodyne Research Inc.), BC was measured by AE33, and NR-$PM_1$ aerosol species were measured by an Aerodyne aerosol chemical speciation monitor (ACSM) (Chen et al., 2015).

Continuous vertical measurements from ground to 240 m (nighttime)/260 m (daytime) were also conducted during 25-26 November and 30 November - 4 December, 2016. The AE33 and PAX were installed in a small container that was able to travel up and down the BMT. In total, 50 vertical profiles of BC, $b_{sca}$ and $b_{abs}$ were obtained (Table S1). The meteorological variables of temperature ($T$), relative humidity (RH), wind speed (WS) and direction (WD) were measured at 15 heights (8, 15, 32, 47, 65, 80, 100, 120, 140, 160, 180, 200, 240, 280 and 320 m) on the BMT. In addition, mixing layer height (MLH) was retrieved from vertical attenuated backscatter coefficients measured by a single-lens ceilometer (CL51, Vaisala, Finland) (Tang et al., 2015;Tang et al., 2016). All data in this study are reported in Beijing local time (UTC + 8 hr).



## 2.2 Inter-comparisons

Aerosol optical properties and BC measurements from different instruments were evaluated through parallel sampling either before or after the campaign. As shown in Fig. 1b, the one-week inter-comparison of the two AE33s shows an excellent agreement in BC measurements (slope = 1.06, $R^2$ = 0.94). To be consistent, the BC measured by AE33$_2$ was scaled by a factor of 1.06 according to that of AE33$_1$. $b_{ext}$ measured by CAPS PM$_{ext}$ was also highly correlated with that measured by CAPS PM$_{ssa}$ ($R^2$=0.99, slope =1.22) (Fig. 1a). Considering that the CAPS PM$_{ssa}$ was not calibrated before this campaign, $b_{ext}$ measured by the CAPS PM$_{ssa}$ was multiplied by a factor of 1.22 in this study. We also compared the light extinction of PM$_1$ and PM$_{2.5}$ during the period of inter-comparison. As indicated in Fig. 1c, the $b_{ext}$ of PM$_{2.5}$ is nearly twice that of PM$_1$ ($R^2$ = 0.93, slope =1.93), indicating that aerosol particles between 1 – 2.5 µm are also important for light extinction in this study. In addition, $b_{sca}$ measured by CAPS PM$_{ssa}$ (630 nm) and PAX (870 nm) is highly correlated ($R^2$=0.98), and the slope of 2.2 suggests a scattering angstrom efficiency (SAE) of approximately 2.4 (Fig. 1d). Figure 2a shows a comparison of BC measured by AE33 and SP-AMS for the entire study. It is clear that both independent measurements are highly correlated ($R^2$=0.98). In our previous study, $b_{abs}$ of BC at 630 nm was derived using a mass absorption cross-section (MAC) of 7.4 m$^2$g$^{-1}$, i.e., $b_{abs}$ = BC × 7.4 (Han et al., 2017). The BC derived $b_{abs}$ is highly correlated with that from the CAPS PM$_{ssa}$ measurements ($R^2$ = 0.89, slope = 1.01), i.e., $b_{abs} = b_{ext} - b_{sca}$. All these results suggest that $b_{abs}$ from different measurements agree reasonably well in this study. We then define the mass absorption cross-section (MAC) of BC at 630 nm as.

$$MAC = b_{abs} / rBC \qquad (1)$$

Where $b_{abs}$ is the absorption coefficient at 630 nm, and $rBC$ is the refractory BC from SP-AMS measurements.

## 2.3 Calculations of single scattering albedo and brown carbon

Although single scattering albedo (SSA, $\lambda$ = 630 nm) of PM$_1$ can be directly calculated from the CAPS PM$_{ssa}$ measurements as SSA = $b_{sca}/b_{ext}$, the SSA of PM$_{2.5}$ at ground level and 260 m are both derived as:



$$SSA = \frac{b_{ext} - b_{abs}}{b_{ext}} = \frac{b_{ext} - 7.4 \times BC}{b_{ext}} \qquad (2)$$

Note that all SSA values discussed below refer to $PM_{2.5}$ at 630 nm unless otherwise stated.

We also estimate the absorption of BrC assuming that BC is the only absorber at 950 nm with an AAE of 1 (Fig. S1). The absorption

coefficient of BC at 370 nm is derived using a fitted power law relationship at seven wavelengths (Ran et al., 2016) (Eq. 3). BrC at

370 nm is then derived by subtracting the BC absorption from the total measured absorption (Eq. 4).

$$b_{abs,370nm,BC} = b_{abs,950nm} \left(\frac{370}{950}\right)^{-AAE} \qquad (3)$$

$$b_{abs,370nm,BrC} = b_{abs,370nm,Total} - b_{abs,370nm,BC} \qquad (4)$$

Where $b_{abs,370nm,BC}$ is the BC absorption at 370 nm, and AAE is 1. Although the absorption of dust is much less than those of BC

and BrC because of much lower MAC (Yang et al., 2009), the data with potential influences of dust as indicated by high Ca

content (Fig. S2) were excluded in calculation of the absorption of BrC at 370 nm.

## 3.Results and discussion

### 3.1 General description

The average $b_{ext}$ (±1σ) were 547.0 ± 555.9 and 387.8 ± 395.2 $Mm^{-1}$ at ground level and 260 m, respectively. Compared with previous

measurements in winter at the same site (Han et al., 2017;Wang et al., 2017b), $b_{ext}$ is much higher in this study due to more frequent

occurrence of severe haze episodes, e.g., six haze episodes in Fig. 3. The results suggest that the PM pollution in winter in Beijing

was still severe although air quality has been continuously improved in recent years. We also note that the average $b_{ext}$ at 260 m was

on average 29.1% lower than that at ground level, suggesting a considerable vertical gradient in $b_{ext}$ in winter.

As shown in Fig. 3, $b_{sca}$ and $b_{abs}$ varied similarly to $b_{ext}$, and the average $b_{sca}$ and $b_{abs}$ at 260 m were 337.6 (±356.0) and 36.6 (±33.9)



Mm$^{-1}$, respectively, which were 26.5% and 22.5% lower than those at ground level (459.5 ±483.1 and 47.2 ±2.4 Mm$^{-1}$, respectively).

Since the SSA is the ratio of $b_{sca}$ and $b_{ext}$, the variations in SSA at ground level were similar to those at 260 m for most of the time, although several periods with much lower SSA at ground level were observed due to large emissions of BC at midnight, e.g., 17 November and 1 December. The average (±1σ) SSA was 0.89 (±0.04) and 0.88 (±0.04) at ground level and 260 m, respectively,

which was overall consistent with those (0.84 – 0.91) in previous studies (Wang et al., 2017b;Han et al., 2017;Han et al., 2015). Figure 3 also shows that high SSA values were typically associated with haze episodes while low values typically occur during clean periods. These results suggest an increasing role of scattering aerosols, e.g., secondary inorganic aerosol species, during haze episodes.

## 3.2 Vertical differences of aerosol optical properties.

All aerosol optical properties including $b_{ext}$, $b_{sca}$, and $b_{abs}$ showed overall similar variations between ground level and 260 m with $R^2$ between ground and 260 m varying from 0.70 to 0.81. However, large vertical differences were also observed during specific periods, for example, the cleaning stages of haze episodes (26 November, 4 and 7 December), and several midnights with largely enhanced BC emissions (17 and 29 November). As indicated in Figs. 4a and 4b, the periods with low values of ratio$_{260m/ground}$ were characterized by correspondingly low MLH (typically less than 300 m) and much higher RH at ground level than 260 m.  Figure 5

further shows that the vertical ratios$_{260m/ground}$ of $b_{sca}$ and $b_{abs}$ varied substantially throughout the study with the values below 1 for most of the time (89% and 84%, respectively). These results suggest that the vertical differences of aerosol optical properties were complex in winter in Beijing, but overall higher heights showed lower values. We also observed different frequencies of the ratio$_{260m/ground}$ between $b_{sca}$ and $b_{abs}$ (Fig. 5). While the ratio$_{260m/ground}$ of $b_{sca}$ varied mainly between 0.6 and 0.8, that of $b_{abs}$ was between 0.8 and 1.0, highlighting the different contributions of scattering and absorbing aerosols at different heights. Although $b_{abs}$

at 260 m was 21% lower than that at ground level (37 vs. 47 Mm$^{-1}$), the relative contribution of absorbing aerosols was relatively higher. One reason is that a large fraction of BC in Beijing is from regional transport (Sun et al., 2016), and the ratio of BC to scattering aerosol species in Beijing surrounding regions is high, likely due to the strict control of heavy-duty/diesel trucks and also



much reduced coal combustion emissions in the city of Beijing.

Figure 6 shows the diurnal cycles of aerosol optical properties at 260 m and ground level after excluding the periods with cleaning processes, while those of entire study are presented in Fig. S3. The diurnal profiles of $b_{sca}$ and $b_{abs}$ were overall similar between ground level and 260 m, which were both characterized by higher values at night and lower values during daytime. Such diurnal

patterns were opposite to those of MLH which showed gradual increases from 300 - 400 m at nighttime to ~600 – 700 m in the late afternoon (Fig. 5a). These results indicate that the variation of PBL height plays an important role in driving the diurnal variations of aerosol optical properties. This is further supported by the diurnal variations in ratios$_{260m/ground}$, and correlations between 260 m and ground ($R^2_{260m\ vs.\ ground}$). As shown in Fig. 6, the ratio$_{260m/ground}$ of $b_{sca}$ increased from ~0.70 at nighttime to ~0.85 during daytime, while that of $b_{abs}$ increased from 0.71 to 0.96 as a result of rising PBL. $R^2$ of $b_{sca}$ and $b_{abs}$ also increased up to > 0.90, suggesting that

scattering and absorbing aerosols were relatively well mixed vertically. Because of the influences of enhanced local primary emissions and the reduced vertical mixing, both ratios and $R^2$ of $b_{abs}$ between 260 m and ground level decreased substantially at nighttime. It should be noted that the ratio$_{260m/ground}$ of $b_{sca}$ did not reach 1 in the late afternoon even when the MLH was above 300 m. These results suggest that other factors also contributed the vertical differences of $b_{sca}$ in addition to MLH. For example, $b_{sca}$ is mainly contributed by secondary aerosols (e.g., ammonium sulfate, ammonium nitrate, and SOA) (Han et al., 2015;Wang et al.,

2015), while the secondary production at different heights could be different due to the different precursor concentrations and oxidants and also different phase partitioning since $T$ falls and RH increases with height. Secondary aerosols are typically formed over a regional scale, which also supports the good correlations of $b_{sca}$ between ground level and 260 m at nighttime.

SSA at ground level showed a clear diurnal trend with the values increasing from 0.85 at 8:00 to 0.89 at 12:00, and then remained at relatively high levels in the late afternoon. Such a diurnal profile is most likely related to photochemical production of secondary

aerosol in the daytime, which is supported by the gradual increase in SSA as a function of photochemical age (-log(NO$_x$/NO$_y$, Fig. S4). Note that the diurnal variation of SSA is relatively smaller than those in autumn and summer at the same site (Han et al., 2015;Han et al., 2017). A possible explanation is that the enhancement of light scattering caused by photochemical production is weaker in winter.  The diurnal profile of SSA at 260 m was relatively similar to that at ground level, but the variations were much



smaller. We also noticed that SSA at ground level was lower than that at 260 m at nighttime due to enhanced light absorption of BC from local emissions. Indeed, such diurnal differences are closely related to composition differences between ground level and 260 m (Zhao et al., 2017).

**3.3 Evolution of vertical profiles of aerosol optical properties.**

50 vertical profiles of $b_{sca}$ and $b_{abs}$ from ground to 260 m (day) or 240 m (night) were obtained in this study (Table S1). As shown in Fig. 5, the vertical ratios of $b_{sca}$ and $b_{abs}$ between ground and 240/260 m agreed well with those determined from simultaneous real-time measurements at the two heights. Although the ratios$_{260/ground}$ of $b_{sca}$ and $b_{abs}$ varied largely from ~0.1 to 1.6, those of SSA showed small vertical differences for most of the time, e.g., < 0.04.   Figure 7 presents the evolution of vertical profiles during a severe haze episode on 2 – 4 December. Before the formation of haze episode, aerosol particles were relatively well mixed vertically,

and the vertical differences of $b_{sca}$ and $b_{abs}$ (M16 and M17) were both less than 30%. This was consistent with the high MLH (~600 m) during this period of time. After M16 (16:00), $b_{ext}$ increased rapidly from less than 100 Mm$^{-1}$ to > 300 Mm$^{-1}$ in four hours associated with a decrease of MLH from ~600 m to ~400 m and a change of air masses from the north to the south-west. We note that $b_{sca}$ and $b_{abs}$ increased simultaneously across different heights, suggesting that both regional transport and the decrease of MLH have played important roles in the formation of this haze episode. However, SSA showed a gradual increase from ~0.82 (M16) to

~0.90 (M19), consistent with the large increases in the contributions of scattering nitrate and SOA during this period (Xu et al., 2018).

The vertical profiles had significant changes at nighttime (M19 and M20) as the occurrence of $T$ inversion. As shown in Fig. 7c, a strong $T$ inversion by approximately 1 °C was observed between 150 – 320 m, and RH was decreased by ~5%. While $b_{sca}$ and $b_{abs}$ were relatively well vertically mixed below the $T$ inversion, they had significant changes in both absolute values and relative

contributions above the layer. In particular, SSA increased considerably from ~0.81 – 0.82 to ~0.88 – 0.90 above the $T$ inversion, suggesting that scattering and absorbing aerosols differed significantly below and above $T$ inversion. In fact, aerosol composition at ground level during M19 and M20 was characterized by higher BC (10%) and organics (48 – 54%) than that at 260 m (8% and 44

– 46%, respectively), and secondary nitrate was correspondingly low (16 – 17% vs. 23%). Because of the $T$ inversion, the enhanced

light absorbing aerosol from local emissions could not be well mixed to 260 m, leading to lower SSA at ground level.  In addition,

the higher mass fraction of scattering secondary inorganic aerosol (SIA) at 260 m could also increase the SSA independently (Fig.

S5).  Air pollution was the severest at the night of 3 December which was associated with a large increase in RH below 200 m,

mostly likely a light fog event, and consistently low MLH (< 200 m). As a result, large vertical differences were observed for both

$b_{sca}$ and $b_{abs}$. In particular, $b_{sca}$ and $b_{abs}$ start to decrease at ~120 m, and the ratios$_{240m/ground}$ decrease rapidly from ~0.8 (M23) to <0.1

(M25).  Such large vertical gradients were also due to the earlier cleaning of air pollutants at higher heights (Fig. 7d).  Although the

vertical changes in $b_{sca}$ and $b_{abs}$ were significant, those of SSA were small, which were due to the small changes in relative

contributions of BC and non-BC aerosols at ground level and 260 m during this period of time (Xu et al., 2018). Our results suggest

that vertical profiles of aerosol optical properties can have significant changes during the formation and evolution of haze episodes

depending on changes in meteorological conditions and source emissions.

### 3.3 MAC of BC.

Figure 9a shows that the MAC of BC increased substantially from 9.5 to 13.2 $m^2 g^{-1}$ as the increase of the mass ratio of non-$r$BC

material to $r$BC in $r$BC-containing particles ($M_R$) from 3.2 to 6.7. Considering that the light absorption of BrC at 630 nm was

relatively small, the increases in MAC could be mainly due to the lensing effect of BC-coated materials. The average $M_R$ is 4.66

(±0.95) indicating that most $r$BC-containing particles contain a large amount of non-$r$BC constituents. According to SP-AMS

measurements, the average composition of $r$BC-containing particles was dominated by organics (59%) and $r$BC (17%), of which

66% of organics are from primary emissions of coal combustion and biomass burning (Wang et al., 2018). This is also consistent

with low O/C ratios (< 0.3) during this study (Fig. 10b). Although the $r$BC-coated materials were dominated by primary aerosol, the

increases in MAC were more associated with secondary aerosol. As shown in Fig. 9b, the contributions of POA (BBOA+FFOA)

decreased from 67% to 16% as $M_R$ increased from 3 to 7, in contrast, the contributions of secondary aerosol species (= SOA+SIA)

increased from 29% to 80%. This was also supported by the correlations between MAC and $M_R$ across different levels. As shown

in Fig. 9a, the increases of MAC as a function of $M_R$ were relatively small, typically less than 20% at low PM levels (< 100 μg m$^{-3}$) when the contribution of POA was much higher than that of secondary aerosol species (37% vs. 59%), while MAC increased significantly by nearly 40% during periods with high PM levels (> 150 μg m$^{-3}$) and higher contribution of secondary aerosol (Fig. S6). We also found that the increases in MAC and $M_R$ were both associated with the corresponding increases in O/C ratios, indicating that photochemical processing plays an important role in changing the MAC and coating of BC in winter. Indeed, MAC showed a pronounced diurnal cycle with a clear daytime increase from ~11 to ~13 m$^2$ g$^{-1}$, (Fig. 10a), and such a diurnal pattern agreed well with that of O/C, an indicator of the extent of chemical aging of organic aerosol. Consistently, MAC was almost linearly correlated with O/C as indicated in Fig. 10b. As O/C increased from 0.1 to 0.25, MAC increased from ~10 to 14 m$^2$ g$^{-1}$. These results suggest that secondary production from photochemical processing can contribute to the light absorption enhancement of BC in daytime by approximately 20%, and up to 40% in this study.

### 3.4 Brown carbon.

As shown in Fig. S7, the average AAE derived from the wavelength-dependent absorption were 1.58 and 1.60 at the ground and at 260 m, respectively, indicating the importance of non-BC light absorbers. Indeed, the absorption of BrC ($b_{abs, BrC}$) on average account for 46% and 48% (Fig. S1) of the total absorption at 370 nm at ground and 260 m, supporting the importance of BrC absorption at ultraviolet wavelength. It should be noted that the extraction of $b_{abs, BrC}$ from total absorption may introduce uncertainty since the AAE of BC is also influenced by BC size and coatings(Liu et al., 2015a). It is interesting to note that the fraction of $b_{abs,BrC}$ in $b_{abs}$ increased from ~0.35 to ~0.5 as the increase of the ratio of organics between SP-AMS and HR-AMS, suggesting that $r$BC-related OA materials were more light absorbing than the total OA. In fact, the organic mass measured by the SP-AMS is dominated by coal combustion and biomass burning emissions (Wang et al., 2018), two important sources of BrC (Yan et al., 2015;Sun et al., 2017;Du et al., 2014). Further supporting evidence is that the mass absorption coefficient of BrC (MAC$_{org, 370 nm}$) showed a pronounced diurnal cycle with higher values at nighttime, consistent with the enhanced nighttime primary emissions in winter. Figure 11a also shows that the nighttime $b_{abs, BrC}$ at ground level was much higher than that at 260 m. One explanation is that a large amount of local primary



emissions at nighttime is not well vertically transported to 260 m. The small late afternoon peak indicates that secondary aerosol from photochemical production can also contribute to the BrC absorption.

$b_{abs, BrC}$ correlated the best with BBOA and FFOA ($R^2$ = 0.83 and 0.81, respectively) and moderately correlated with photochemical SOA ($R^2$ = 0.46) at ground level. These results suggest that biomass burning, coal combustion, photochemical production are three major sources of BrC in winter. We then estimated the contributions of different OA sources to BrC using (1) a multiple linear regression model as indicated by Eq. (5) and PMF with OA factors and BrC absorption as input.

$$b_{abs, BrC} = m_1 \times [FFOA] + m_2 \times [COA] + m_3 \times [BBOA] + m_4 \times [OOA] + m_5 \times [OPOA] + m_6 \times [aq\text{-}OOA] \qquad (5)$$

The results are very consistent between the two approaches which both show that coal combustion is the major source contribution of BrC at ground level, on average accounting for 48 – 55% (Fig. 12). The contributions of BBOA and OOA to BrC were comparable, which were 17% and 19 – 20%, respectively. In comparison, cooking emissions and aqueous-phase processing tend to be minor contributions of BrC in winter in Beijing. Our results are consistent with a recent study which highlighted the importance of fossil source contribution to BrC in winter in Beijing (Yan et al., 2017). Also, large contributions of BBOA and OOA to BrC were also observed in Switzerland during winter (Moschos et al., 2018).

**4 Conclusions**

We conducted comprehensive measurements of aerosol optical properties at both ground level and 260 m in winter in Beijing using a suite of state-of-the-art instruments, e.g., two CAPSs, two AE33s, SP-AMS, and PAX. The inter-comparisons showed excellent agreements among different instruments. $b_{ext}$, $b_{sca}$, and $b_{abs}$ varied dynamically throughout the campaign, but overall showed similar trends between ground level and 260 m. On average, $b_{sca}$ and $b_{abs}$ were 337.6 and 36.6 Mm$^{-1}$ at 260 m, which are 26.5% and 22.5% lower than those at ground level (459.5 and 47.2 Mm$^{-1}$, respectively). In fact, ratios$_{260m/ground}$ of $b_{sca}$ and $b_{abs}$ varied substantially from <0.1 to ~1.5, indicating very complex vertical differences of aerosol optical properties in winter in Beijing. In particular, low ratios$_{260m/ground}$ were frequently observed during cleaning stages of severe haze episodes and nighttime with strong local emissions

and low MLH. The diurnal cycles of ratios$_{260m/ground}$ and $R^2_{260m/ground}$ further indicated that the change of MLH played an important role in driving the vertical differences between daytime and nighttime. Although the vertical difference of SSA was not as much as those of $b_{sca}$ and $b_{abs}$, we observed ubiquitously higher values at ground level than 260 m, suggesting relative more scattering aerosols at ground site. A case study further showed complex vertical changes of aerosol optical properties during the formation, evolution

and cleaning stages of the haze episode, which were mainly due to the changes in meteorological conditions (e.g., $T$ inversion), MLH, and source emissions.

The MAC of BC showed a wide range from 9.5 to 13.2 $m^2\,g^{-1}$. MAC was observed to increase substantially as a function of $M_R$ during periods with high PM levels, and the increases were closely related with the increases in secondary aerosols species. The positive relationship between MAC and O/C, and their diurnal profiles further suggest that photochemical processing can increase

MAC by approximately 20%, and up to 40%. It should be noted that the MAC measured in this study could be different from the MAC of ambient aerosol since the particles were dried and sampled at constant room temperature. BrC was a large contributor to the absorption at 370 nm in winter, on average accounting for 46% and 48% at ground level and 260 m, respectively. By linking with OA factors from PMF analysis, we found that BrC in winter is dominantly contributed by coal combustion (48 – 55%) followed by biomass burning (17%) and photochemical SOA (~20%).

***Data availability.*** The data in this study are available from the authors upon request (sunyele@mail.iap.ac.cn).

***Competing interests.*** The authors declare that they have no conflict of interest.

***Acknowledgements.*** This work was supported by the National Natural Science Foundation of China (91744207, 41571130034, 41575120) and the National Key Project of Basic Research (2014CB447900). Qingqing Wang acknowledges the support from the General Financial Grant from the China Postdoctoral Science Foundation (2017M610972) and the National Postdoctoral Program

for Innovative Talents (BX201600157).



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



**Table 1.** A summary of measurements in this study

|  | Instruments | Manufacturer | Properties measured | Resolution |
|---|---|---|---|---|
| Ground | Cavity attenuated phase shift single scattering albedo monitor (CAPS PM$_{ssa}$) | Aerodyne Research, Inc. | PM$_1$ $b_{ext}$ and $b_{sca}$ ($\lambda$= 630 nm) | 1 s |
|  | Aethalometer (AE33$_1$) | Magee Scientific Corp. | BC (370, 470, 520, 590, 660, 880, and 950 nm) | 1 min |
|  | Photoacoustic Extinctiometer (PAX) | Droplet Measurement Technologies | PM$_{2.5}$ $b_{ext}$ and $b_{abs}$ ($\lambda$= 870 nm) | 1 s |
|  | High-Resolution Time-of-Flight Aerosol Mass Spectrometer (HR-AMS) | Aerodyne Research, Inc. | NR-PM$_1$ species (organics, sulfate, nitrate, ammonium, and chloride) | 5 min |
|  | Soot Particle Aerosol Mass Spectrometer (SP-AMS) | Aerodyne Research, Inc. | BC-containing PM$_1$ species ($r$BC, organics, sulfate, nitrate, ammonium, and chloride) | 5 min |
| 260 m | Cavity Attenuated Phase Shift Particulate Matter extinction monitor (CAPS PM$_{ext}$ | Aerodyne Research, Inc. | PM$_{2.5}$ $b_{ext}$ ($\lambda$=630 nm) | 1 s |
|  | Aethalometer (AE33$_2$) | Magee Scientific Corp. | BC (370, 470, 520, 590, 660, 880, and 950 nm) | 1 min |
|  | Aerosol Chemical Species Monitor (ACSM, Aerodyne) | Aerodyne Research, Inc. | NR-PM$_1$ species (organics, sulfate, nitrate, ammonium, and chloride) | 5 min |
| Vertical | Photoacoustic Extinctiometer (PAX) | Droplet Measurement Technologies | PM$_{2.5}$ $b_{sca}$ and $b_{abs}$ ($\lambda$=870 nm) | 1 s |
|  | Aethalometer (AE33$_1$) | Magee Scientific Corp. | BC (370, 470, 520, 590, 660, 880, and 950 nm) | 1 s |

**Table 2.** A summary of average (±1σ) meteorological parameters and optical properties at ground level and 260 m during this study

|  | 260 m | Ground | $R_{260m/Ground}$ |
|---|---|---|---|
|  | *Meteorological parameters* | | |
| $T$ (°C) | 2.3 ± 3.7 | 4.3 ± 3.7 | 0.53 |
| RH (%) | 49.1 ± 23.5 | 48.5 ± 20.7 | 1.01 |
| WS (m s$^{-1}$) | 4.0 ± 2.4 | 1.4 ± 1.0 | 2.85 |
|  | *Optical properties* | | |
| $b_{ext}$ (Mm$^{-1}$) | 387.8 ± 395.2 | 547.0 ± 555.9 | 0.71 |
| $b_{sca}$ (Mm$^{-1}$) | 337.6 ± 356.0 | 459.5 ± 483.1 | 0.73 |
| $b_{abs}$ (Mm$^{-1}$) | 36.6 ± 33.9 | 47.2 ± 2.4 | 0.78 |
| SSA | 0.88 ± 0.04 | 0.89 ± 0.04 | 0.98 |
| AAE | 1.60 ± 0.18 | 1.58 ± 0.15 | 1.01 |





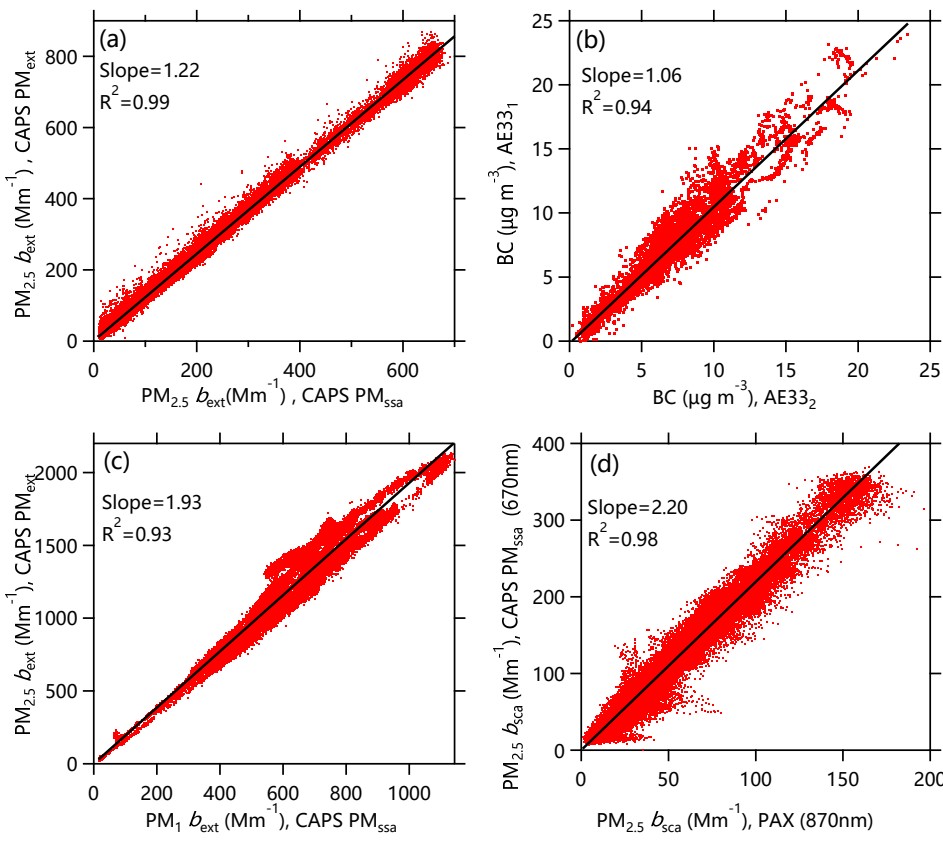

**Figure 1. Inter-comparisons of aerosol optical properties and BC: (a) $b_{ext}$ of PM2.5 from CAPS PM$_{ext}$ and CAPS PM$_{ssa}$, (b) BC from two AE33s, (c) $b_{ext}$ of PM2.5 from CAPS PM$_{ext}$ vs. $b_{ext}$ of PM1 from CAPS PM$_{ssa}$, (d) $b_{sca}$ from CAPS PM$_{ssa}$ vs. $b_{sca}$ from PAX.**





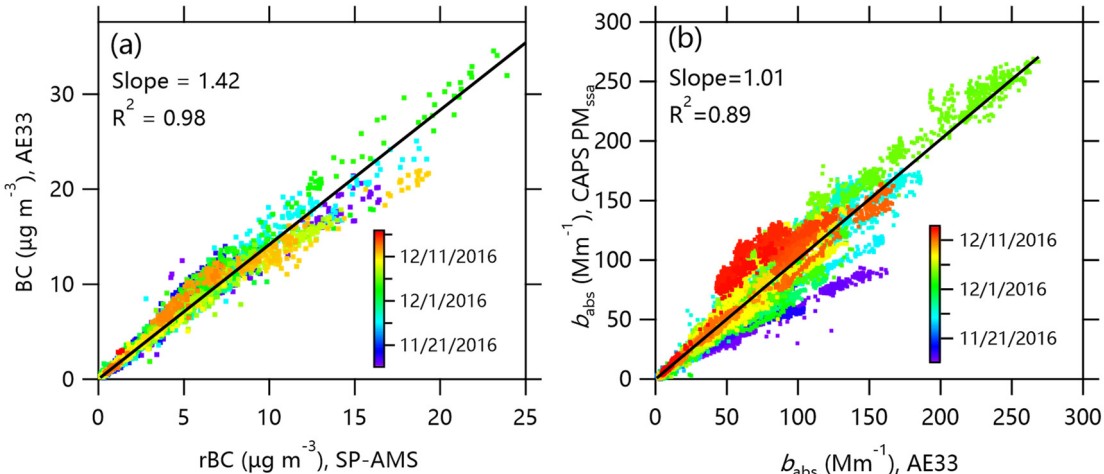

**Figure 2. Comparisons of (a) BC measurements between AE33 and SP-AMS, (b) $b_{abs}$ calculated from the difference of $b_{ext}$ and $b_{sca}$ of CAPS PM$_{ssa}$ and that was derived from BC.**





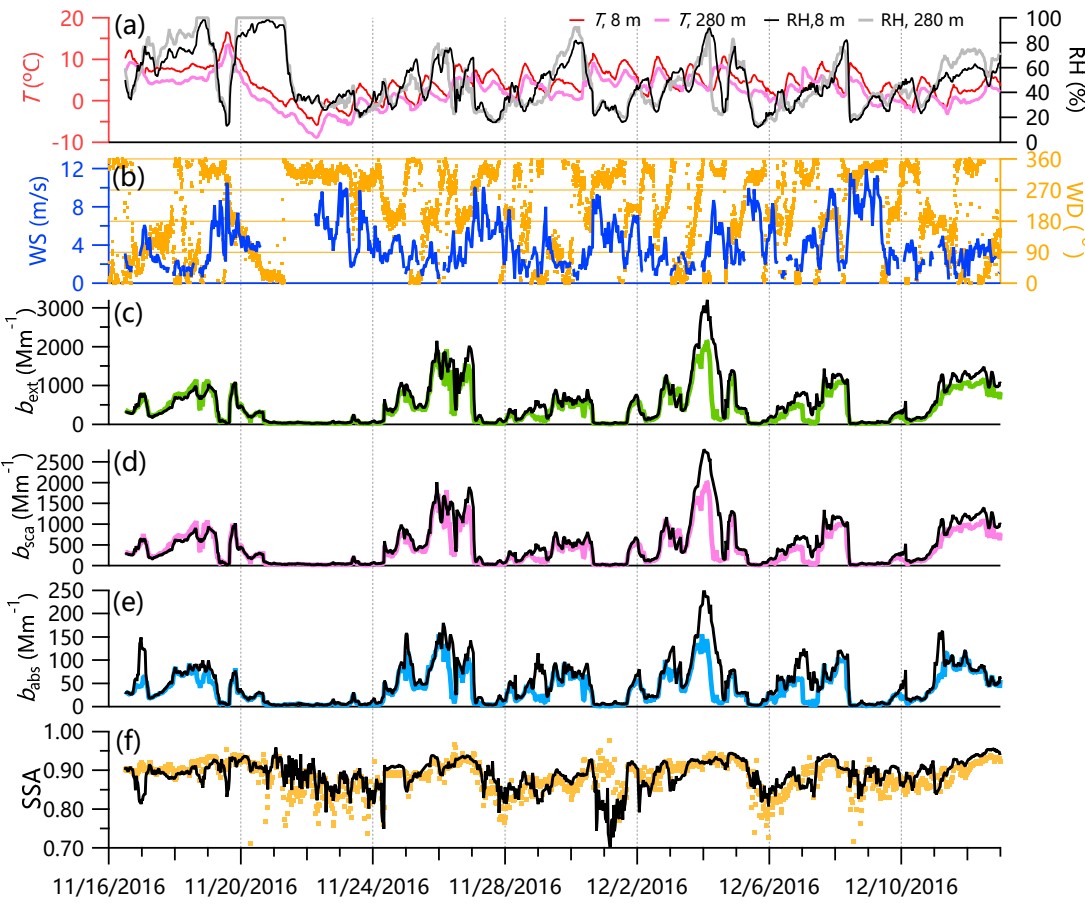

**Figure 3. Temporal variations of (a) temperature ($T$) and relative humidity (RH) at ground level and 280 m, (b) wind speed (WS) and wind direction (WD) at 280 m, and comparisons of the time series of optical properties of PM$_{2.5}$ between ground level (black lines) and 260 m (color lines): (c) $b_{ext}$, (d) scattering coefficient, (e) absorption coefficient, (f) SSA.**

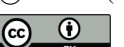


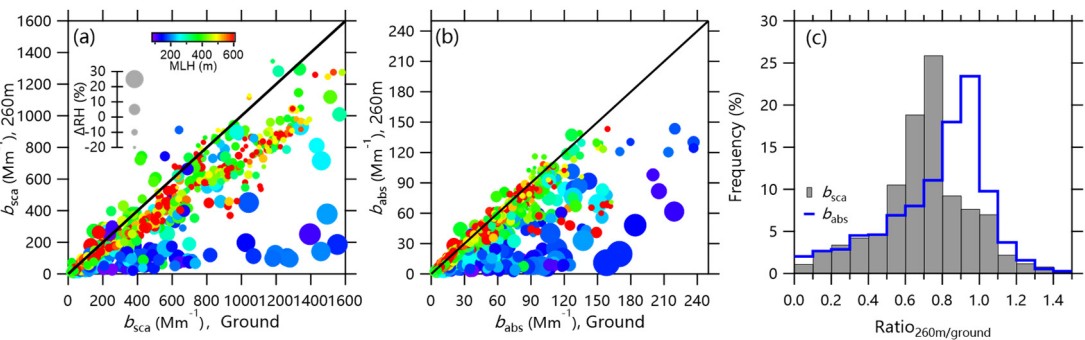

**Figure 4. Comparisons of (a) $b_{sca}$ and (b) $b_{abs}$ between 260 m and ground level. The scatter plot is color coded by mixed layer height, and the marker size is proportional to the RH difference between ground level and 280 m. (c) shows the frequencies of ratios of 260 m to ground level for $b_{sca}$ and $b_{abs}$.**

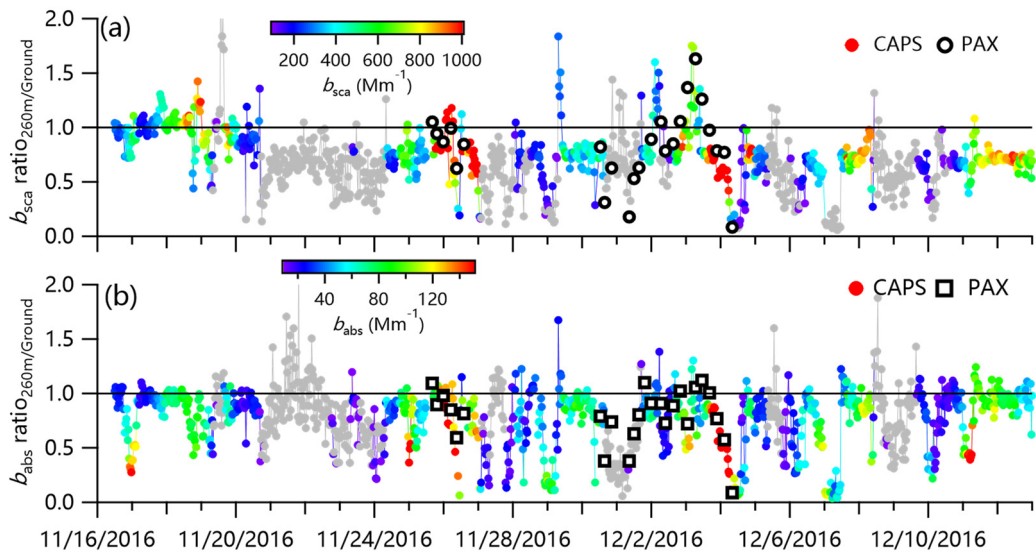

**Figure 5. Time series of hourly aveage ratios$_{260m/ground}$ for bsca and babs during this study. The ratios are color coded by $b_{ext}$, and the black circles and squares are those from continuous vertical measurements with PAX.**

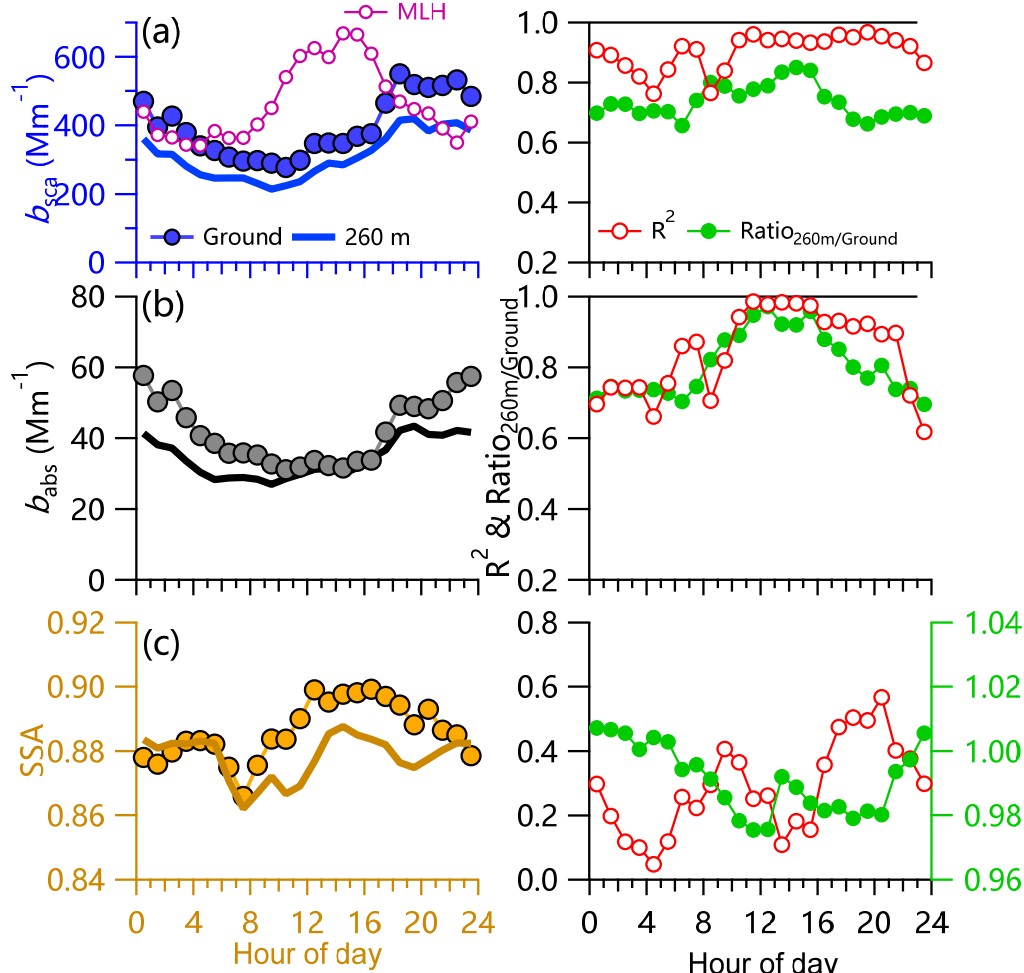

**Figure 6.** Diurnal cycles of (a) $b_{sca}$, (b) $b_{abs}$, and (c) SSA at 260 m and ground level for the entire study. Right panel shows the diurnal cycles

of correlation coefficients ($R^2$) and ratio$_{260m/ground}$ for $b_{sca}$, $b_{abs}$, and SSA. Note that the diurnal cycles above excluded 5 periods with cleaning

processes (November 19 1:00 to 9:00, November 26 1:00 to 12:00, November 26 19:00 to November 27 3:00, December 3 20:00 to December

5    4 16:00, and December 17 00:00 to 12:00) while those of entire study are shown in Fig. S3.





**Figure 7. Evolution of vertical profiles of $b_{sca}$, $b_{abs}$ and SSA during 2 – 4 December. Also shown are (a-d) meteorological variable of $T$, RH, WD, and WS, and (e) time series of $b_{ext}$ at ground and MLH. M16-M25 refer to the number of vertical profiles.**





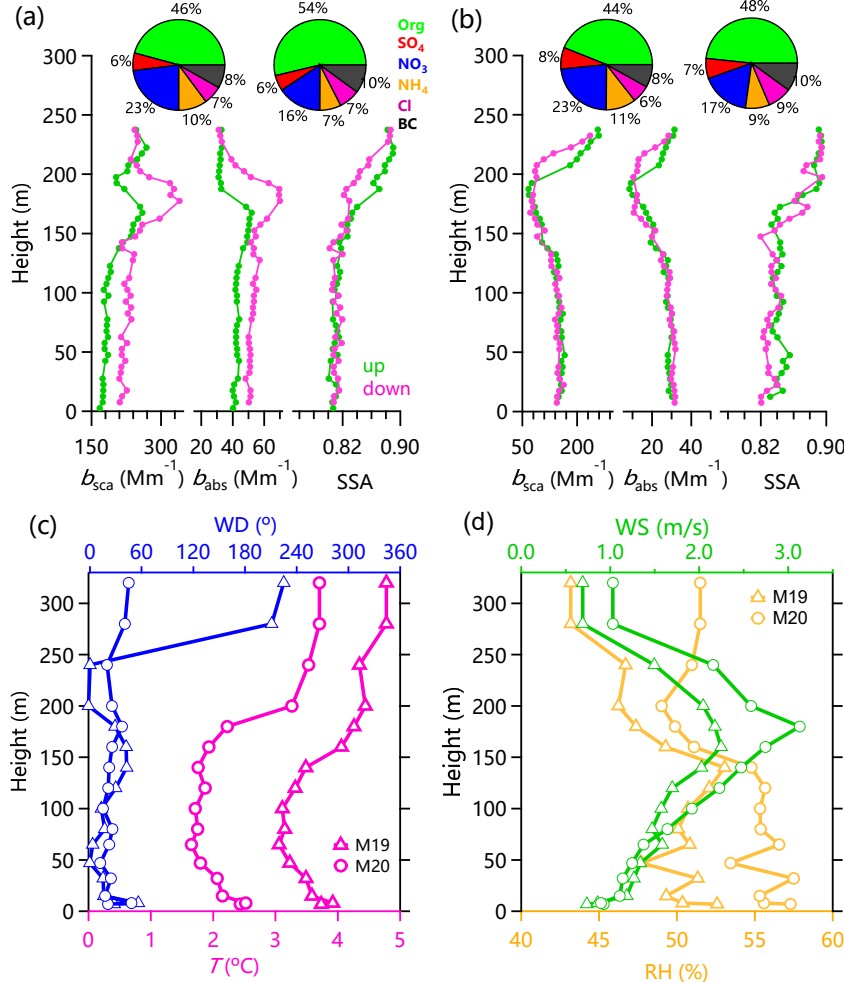

**Figure 8. Vertical profiles of $b_{sca}$, $b_{abs}$ and SSA measured by PAX at 880 nm during (a) M19 and (b) M20. (c) and (d) show the average vertical profiles of T, RH, WS, and WD during M19 and M20, respectively. The pie charts show average chemical composition of PM$_1$ at 260 m (left) and ground level (right) during M19 and M20, respectively.**



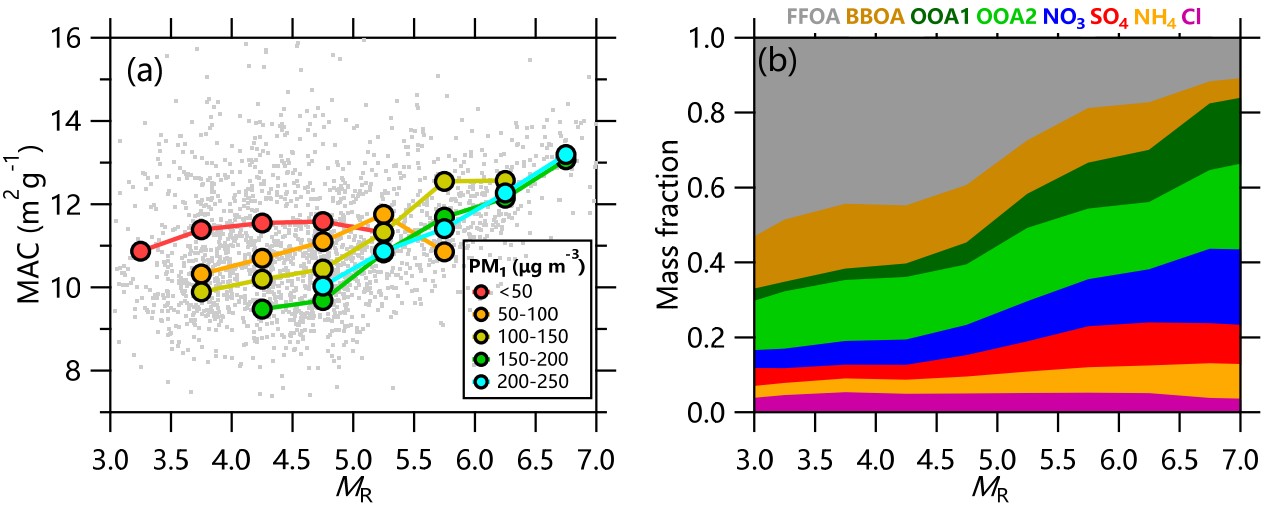

**Figure 9. (a) Variations of mass absorption cross-section (MAC) as a function of $M_R$ at different pollution levels and (b) variations of mass fraction of $r$BC-coated species as a function of $M_R$.**

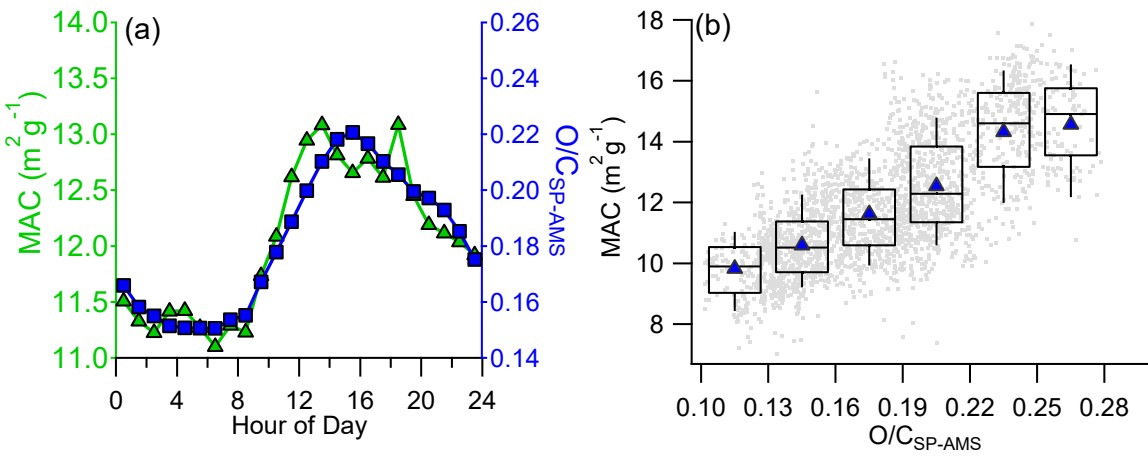

**Figure 10. (a) Diurnal cycles of MAC and O/C ratio of SP-AMS (O/C$_{SP-AMS}$), and (b) variations of MAC as a function of O/C$_{SP-AMS}$. The data are binned according to O/C. The median (horizontal line), mean (triangle), 25th and 75th percentiles (lower and upper box), and 10th and 90th percentiles (lower and upper whiskers) are also shown.**



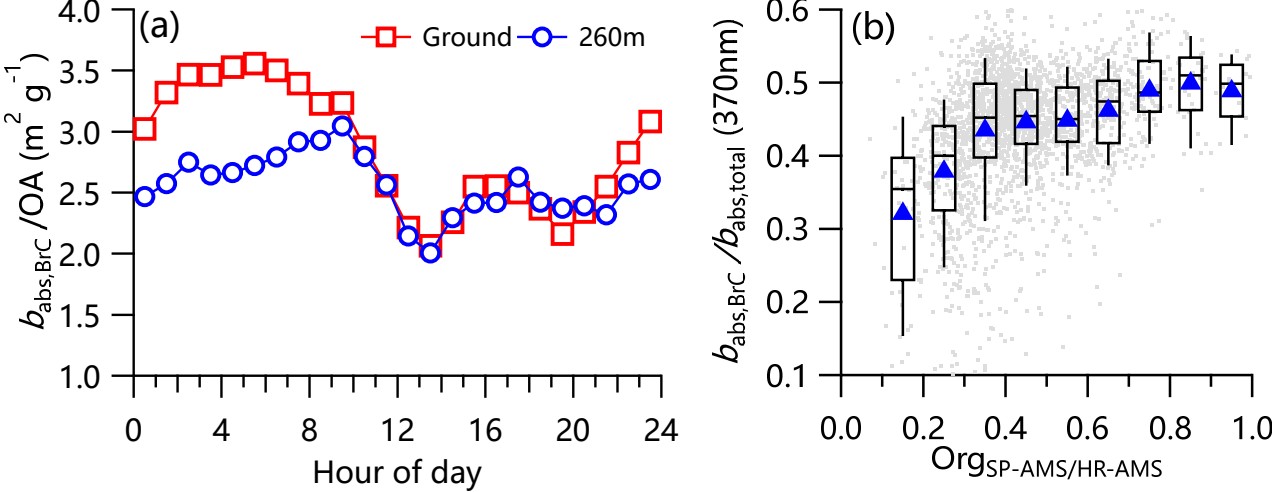

**Figure 11. (a) Diurnal cycles of $b_{abs,BrC}$/OA, i.e., MAC$_{org, 370 nm}$ at ground level and 260 m, (b) variations of the ratio of BrC absorption to the total absorption at 370 nm as a function of the ratio of organics between SP-AMS and HR-AMS. The data are binned according to O/C. The median (horizontal line), mean (triangle), 25th and 75th percentiles (lower and upper box), and 10th and 90th percentiles (lower and upper whiskers) are also shown.**

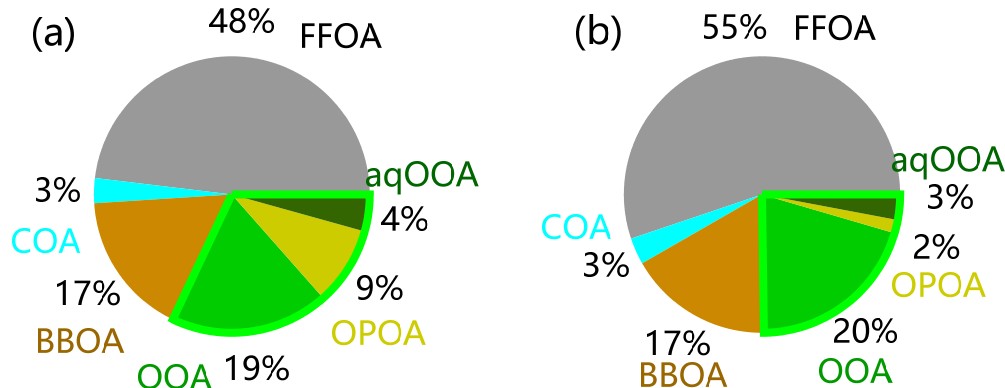

**Figure 12. Average contributions of OA sources to BrC absorption at ground level from (a) linear regression model analysis, and (b) positive matrix factorization (Fig. S8). The sources of OA include fossil fuel-related OA (FFOA), cooking OA(COA), biomass burning OA (BBOA), oxygenated OA (OOA), oxidized primary OA (OPOA), aqueous-phase OOA (aq-OOA) (Xu et al. 2018).**