# Peer review of "Vertical characterization of aerosol optical properties and brown carbon in winter in urban Beijing, China"

_Atmospheric Chemistry and Physics, 2018_

## Referee Comment (RC1) · Anonymous Referee #1 · 1 Sep 2018

This manuscript studied aerosol optical properties at ground level and at 260 m on a meteorological tower in Beijing in the winter. The findings of this paper were mainly that 1) the mass absorption cross-section MAC of BC ($\lambda$ = 630 nm) was strongly associated with the mass ratio of non-refractory BC materials to refractory BC; 2) the MAC of BC increased with the formation of secondary aerosols; 3) brown carbon was a major component to the total absorption. I recommend publication with reference to the following comments.

1. BrC was determined based on the optical property of aerosols, i.e. the difference between the total absorption and the absorption by BC at 370 nm. Then the absorption

by BrC was apportioned to the OA factors from PMF analysis of AMS data. However, this paper did not discuss the compositions of the BrC. Since HR-AMS can measure the molecular fragments, the authors should also discuss the evidence of BrC from the molecular composition.

2. The authors measured 50 vertical profiles of bsca and babs, but just presented 10 profiles and discuss several ones. Did the other profiles have the same evolution with the discussed ones? The authors should presented all the aerosol optical property profiles and the meteorological parameters, at least in the Supplementary materials.

3. Line 4 in page2: The abbreviation of non-refractory BC as rBC will cause confusion. Rewrite this sentence.

---

## Short Comment (SC1) · 11 Sep 2018

This is a very brief comment to recommend that the authors use the terminology established in Petzold et al. (2013) when referring to aethalometer measurements. That is, use eBC rather than just BC.

Additionally, I would suggest writing MAC(637nm) rather than simply MAC on the figures, as was done in the abstract.

Best regards,

Joel Corbin.

[Figure]

Petzold, A., Ogren, J. A., Fiebig, M., Laj, P., Li, S.-M., Baltensperger, U., Holzer-Popp, T., Kinne, S., Pappalardo, G., Sugimoto, N., Wehrli, C., Wiedensohler, A., and Zhang, X.-Y.: Recommendations for reporting "black carbon" measurements, Atmos. Chem. Phys., 13, 8365-8379, https://doi.org/10.5194/acp-13-8365-2013, 2013.
* * *

---

## Referee Comment (RC2) · Anonymous Referee #2 · 25 Sep 2018

This work did a very nice job in measuring the aerosol optical properties at both ground and 260 m in winter Beijing. The quality of the measurement data were ensured by inter-comparisons across different instruments. The authors found substantial vertical differences in aerosol optical properties (e.g., SSA) at nighttime due to strong local emissions. They also clarified that the lensing effect played an important role in enhancing BC absorption in Beijing. Source apportionment of BrC absorption at 370 nm were conducted using two statistical tools, both suggesting that fossil fuel combustion is the dominant BrC contributor in urban Beijing. This work is very well designed and written, I would recommend this work to be published on Atmos. Chem. Phys. with only one comment to be addressed

The comment is about the method for $b_{abs,270nm,BrC}$ estimation with Eqs (3) and (4).
In Eq (3), the BC absorption at 370nm is estimated using a fitted power law, and BrC at 370 nm is then derived by subtracting the BC absorption from the total measured absorption.
However, this work has demonstrated that the coating of organic aerosol on BC (or "lensing effect") can contribute to the light absorption enhancement of BC substantially (20~40%). In Eqs (3) and (4), such coating effect was not considered, which might lead to an overestimation of $b_{abs,270nm,BrC}$. So please provide an estimation on the uncertainty associated with $b_{abs,270nm,BrC}$ estimation due to OA coatings, or more discussions on the uncertainties.

---

## Author Comment (AC1) · 29 Nov 2018

We are thankful to the three referees for their thoughtful and constructive comments which help improve the manuscript substantially. Following the reviewers' suggestions, we have revised the manuscript accordingly. Listed below are our point-by-point responses in blue to each comment that is repeated in italic.

**Response to Reviewer #1**

*BrC was determined based on the optical property of aerosols, i.e. the difference between the total absorption and the absorption by BC at 370 nm. Then the absorption by BrC was apportioned to the OA factors from PMF analysis of AMS data. However, this paper did not discuss the compositions of the BrC. Since HR-AMS can measure the molecular fragments, the authors should also discuss the evidence of BrC from the molecular composition.*

We thank the reviewer's comments. Molecular characterization of BrC is very important. Unfortunately, HR-AMS uses 70 eV electron impact for ionization, and the molecular information of organic compounds is difficult to retain. Therefore, we compared BrC with OA factors to investigate the potential sources of BrC in this study, and found that coal combustion and biomass burning were two major sources. Although we didn't have molecular information of BrC, previous studies have shown that nitrophenols and aromatic carbonyls are often the major BrC species (Desyaterik et al., 2013;Laskin et al., 2015).

*Specific comments:*

1. *The authors measured 50 vertical profiles of $b_{sca}$ and $b_{abs}$, but just presented 10 profiles and discuss several ones. Did the other profiles have the same evolution with the discussed ones? The authors should present all the aerosol optical property profiles and the meteorological parameters, at least in the Supplementary materials.*

We thank the reviewer's suggestions. In the revised manuscript, we added the rest profiles in Figure S5 in supplementary materials. As shown in Figure S5, most vertical profiles are overall similar to those discussed in the text.

[Figure]

Figure S5. Evolution of vertical profiles of $b_{sca}$, $b_{abs}$ and SSA during 25-26 November (top) and 30 November – 2 December (bottom). Also shown are (a-d) meteorological variable of $T$, RH, WD, and WS, and (e) time series of $b_{ext}$ at ground and MLH. M16-M25 refer to the number of vertical profiles.

*Line 4 in page2: The abbreviation of non-refractory BC as rBC will cause confusion. Rewrite this sentence.*

Revised.

**Response to Reviewer #2**

*This work did a very nice job in measuring the aerosol optical properties at both ground and 260 m in winter Beijing. The quality of the measurement data were ensured by inter-comparisons across different instruments. The authors found substantial vertical differences in aerosol optical properties (e.g., SSA) at nighttime due to strong local emissions. They also clarified that the lensing effect played an important role in enhancing BC absorption in Beijing. Source apportionment of BrC absorption at 370 nm were conducted using two statistical tools, both suggesting that fossil fuel combustion is the dominant BrC contributor in urban Beijing. This work is very well designed and written, I would recommend this work to be published on Atmos. Chem. Phys. with only one comment to be addressed*

We thank the reviewer's positive comments.

*The comment is about the method for $b_{abs,370nm,BrC}$ estimation with Eqs (3) and (4). In Eq (3), the BC absorption at 370nm is estimated using a fitted power law, and BrC at 370 nm is then derived by subtracting the BC absorption from the total measured absorption. However, this work has demonstrated that the coating of organic aerosol on BC (or "lensing effect") can contribute to the light absorption enhancement of BC substantially (20~40%). In Eqs (3) and (4), such coating effect was not considered, which might lead to an overestimation of $b_{abs,370nm,BrC}$. So please provide an estimation on the uncertainty associated with $b_{abs,370nm,BrC}$ estimation due to OA coatings, or more discussions on the uncertainties.*

This is a very good point. Previous studies by Liu et al. (2017) found that the absorption enhancement was small at $M_R <$ 3. Assuming no absorption enhancement at $M_R$ = 3 in this study, the increase in mass absorption cross-section (MAC) of BC due to coating materials was approximately 10 − 20%, and can be up to ~30% when $M_R$ was above 6.5 (Fig. 9). Therefore, the upper limit of absorption enhancement of BC due to "lensing effect" in this study was ~30%, and mostly likely between 10% and 20%. With this, the $b_{abs,370nm,BrC}$ could be overestimated by ~10 − 20%, but should be less than 30%.

Following the reviewer's comments, we added the uncertainties for estimation of BrC in the revised manuscript.

"It should be noted that we might overestimate $b_{\mathrm{abs,370nm,BrC}}$ by approximately 10 − 20% if the contribution of "lensing effect" on BC absorption at $M_R$ = 3 was negligible (Liu et al., 2017)."

**Response to J. C. Corbin**

*This is a very brief comment to recommend that the authors use the terminology established in Petzold et al. (2013) when referring to aethalometer measurements. That is, use eBC rather than just BC.Additionally, I would suggest writing MAC(637nm) rather than simply MAC on the figures, as was done in the abstract.*

*Best regards,*

*Joel Corbin.*

*Petzold, A., Ogren, J. A., Fiebig, M., Laj, P., Li, S.-M., Baltensperger, U., Holzer-Popp, T., Kinne, S., Pappalardo, G., Sugimoto, N., Wehrli, C., Wiedensohler, A., and Zhang, X.-Y.: Recommendations for reporting "black carbon" measurements, Atmos. Chem. Phys., 13, 8365-8379, https://doi.org/10.5194/acp-13-8365-2013, 2013.*

We thank Dr. Corbin's good suggestions. We changed "BC" to "eBC" in the revised manuscript, and included the wavelength for MAC.

References:

Desyaterik, Y., Sun, Y., Shen, X., Lee, T., Wang, X., Wang, T., and Jeffrey Collett, J.: Speciation of "brown" carbon in cloud water impacted by agricultural biomass burning in Eastern China, J. Geophys. Res., 118, 7389–7399, doi:10.1002/jgrd.50561, 2013.

Lack, D. A., and Cappa, C. D.: Impact of brown and clear carbon on light absorption enhancement, single scatter albedo and absorption wavelength dependence of black carbon, Atmospheric Chemistry and Physics, 10, 4207-4220, 10.5194/acp-10-4207-2010, 2010.

Laskin, A., Laskin, J., and Nizkorodov, S. A.: Chemistry of Atmospheric Brown Carbon, Chem. Rev., 10.1021/cr5006167, 2015.

Liu, D., Whitehead, J., Alfarra, M. R., Reyes-Villegas, E., Spracklen, D. V., Reddington, C. L., Kong, S., Williams, P. I., Ting, Y.-C., Haslett, S., Taylor, J. W., Flynn, M. J., Morgan, W. T., McFiggans, G., Coe, H., and Allan, J. D.: Black-carbon absorption enhancement in the atmosphere determined by particle mixing state, Nature Geosci, advance online publication, 10.1038/ngeo2901, 2017.